# A bio-inspired coordination polymer as outstanding water oxidation catalyst via second coordination sphere engineering

Wenlong Li[1,3], Fusheng Li [1,3]*, Hao Yang[1], Xiujuan Wu[1], Peili Zhang[1], Yu Shan[1] & Licheng Sun [1,2]*

First-row transition metal-based catalysts have been developed for the oxygen evolution reaction (OER) during the past years, however, such catalysts typically operate at over-potentials ($\eta$) significantly above thermodynamic requirements. Here, we report an iron/nickel terephthalate coordination polymer on nickel form (**NiFeCP/NF**) as catalyst for OER, in which both coordinated and uncoordinated carboxylates were maintained after electrolysis. **NiFeCP/NF** exhibits outstanding electro-catalytic OER activity with a low overpotential of 188 mV at 10 mA cm$^{-2}$ in 1.0 KOH, with a small Tafel slope and excellent stability. The pH-independent OER activity of **NiFeCP/NF** on the reversible hydrogen electrode scale suggests that a concerted proton-coupled electron transfer (c-PET) process is the rate-determining step (RDS) during water oxidation. Deuterium kinetic isotope effects, proton inventory studies and atom-proton-transfer measurements indicate that the uncoordinated carboxylates are serving as the proton transfer relays, with a similar function as amino acid residues in photosystem II (PSII), accelerating the proton-transfer rate.

[1] State Key Laboratory of Fine Chemicals, Institute of Artificial Photosynthesis, DUT-KTH Joint Education and Research Centre on Molecular Devices, Institute for Energy Science and Technology, Dalian University of Technology, 116024 Dalian, China. [2] Department of Chemistry, School of Engineering Sciences in Chemistry, Biotechnology and Health, KTH Royal Institute of Technology, Stockholm 10044, Sweden. [3]These authors contributed equally: Wenlong Li, Fusheng Li *email: fusheng@dlut.edu.cn; sunlc@dlut.edu.cn

The growing global demand for energy, coupled with an increasing awareness of climate change, has motivated the development of new renewable energy conversion technologies[1,2]. Electrochemical water splitting to produce sustainable $H_2$ fuel has been widely regarded as one of the most promising strategies for energy storage. The efficiency of water splitting is severely hampered by a sluggish process, that is, the oxygen evolution reaction (OER). The multi-electron and proton-coupled OER is a kinetically slow reaction, and as a result, large overpotentials are usually required, which decreases the energy conversion efficiency and hinders the practical applications of devices.

So far, significant efforts have been devoted to fabricate efficient, alternative OER catalysts. Hitherto, $IrO_2$ and $RuO_2$ are two benchmark OER catalysts employed for acidic electrolysis, on account of their high activity for water oxidation. However, the scarcity of Ir and Ru limits their large-scale application[3]. At the same time, extensive efforts have been taken to explore non-noble metal catalysts with outstanding activity and stability for alkaline electrolysis. The bimetallic Fe-Ni composites are regarded as one of the most promising alternatives in the family of OER catalysts, however, such composites operate at high $\eta$ toward the OER and the mechanistic details for the associated elemental steps are poorly understood. Because proton has a much larger mass than that of electron, slow proton/fast electron transfer will be realized[4,5], and the rate of ferrying proton may determine the overall water oxidation reaction. Thence, a comprehensive understanding of the underlying mechanism of proton-coupled interfacial electron transfer process in the rate-determining step (RDS) is vital importance for obtaining efficient OER catalysts, and a new thinking and related design of the surface active site at molecular level to accelerate the sluggish kinetics are urgently needed.

In green plants and cyanobacterial, light driven water oxidation by the oxygen-evolving complex (OEC) in PSII occurs with a high turnover frequency of $100-400 \, s^{-1}$ in vivo at low overpotential ($\eta < 200$ mV)[6,7]. The evolution of oxygen by OEC is an integrated process, not only involving the $Mn_4CaO_5$ cluster as catalyst, but also is assisted by other functional groups in the first and second coordination spheres, such as carboxylate and imidazole ligands from amino-acid residues of the protein backbone[8,9]. These electron-rich ligands strongly stabilize the high valent states of the $Mn_4CaO_5$ cluster and play vital roles in effective water oxidation with low overpotential[7,10]. Inspired by PSII, strong electron donating ligands, such as phenolate and carboxylate ligands are extensively utilized in the structures of homogeneous molecular water oxidation catalysts. Generally, the redox potentials of metal complexes can be tuned by ligand modification, and the incorporation of negatively-charged ligands into metal complexes has previously been shown to stabilize metal centers at higher oxidation states and result in complexes with significantly lower redox potentials[7,10].

Furthermore, the presence of carboxylate ligands in the OEC is crucial for transporting the generated protons during water oxidation. For example, the amino acid residue of Asp170 has been proposed to form a hydrogen bond with a coordinated water molecule on $Mn_4CaO_5$ cluster[11,12], and may act as an internal base for proton transfer during the O-O bond formation[12,13]. Asp61 is also proposed serving the similar function[14,15]. This internal base strategy has been reported when designing molecular water oxidation catalysts, in which carboxylate or pyridine groups are employed as the internal base (proton transfer relay) for proton transfer during water oxidation[16-18]. These second-coordination-sphere effects can be also introduced to the heterogeneous OER catalysts.

Inspired by the OEC structure in PSII, herein, we report a binder-free Ni-Fe coordination polymer (**NiFeCP**) prepared via an in situ electrochemical deposition method on a Ni foam (NF) as catalyst (**NiFeCP/NF**) for OER (Fig. 1). Negatively-charged carboxylate ligands were simultaneously introduced into the **NiFeCP** composites in both coordinated and uncoordinated forms, with the former expected to stabilize the high valence states of the metal centers, and the latter expected to serve as proton transfer relays in the second-coordination-sphere of the active site. To our delight, the as prepared **NiFeCP/NF** electrode exhibits excellent catalytic activity with a low onset potential of 1.41 V (vs. reversible hydrogen electrode, RHE), and a constantly low $\eta$ of 188 mV to reach $10 \, mA \, cm^{-2}$ in 1.0 M KOH electrolyte. The pH-independence OER property combining with deuterium kinetic isotope effects, proton inventory and atom proton transfer studies demonstrates that the uncoordinated carboxylate groups in **NiFeCP** participate in the water oxidation reaction by serving as the proton transfer relays and contribute to the superior activity of the **NiFeCP/NF** catalyst toward OER.

## Results

The **NiFeCP/NF** electrode was prepared via a repeated double-current pulse chronopotentiometry (Supplementary Fig. 1). As a result of the fast reaction process during deposition, electro-deposited materials are often amorphous or possess low crystallinity, which provide opportunities to obtain a polymer comprising both coordinated and uncoordinated carboxylate ligands. The $Fe^{3+}$ incorporated in Ni-based catalysts has been reported to be the key factor for high catalytic performance, although its exact role remains ambiguous[19,20]. The OER electrocatalytic performances of the **NiFeCP/NF** prepared with different Ni:Fe ratios in the electrodeposition solution were measured, as presented in the linear sweep voltammograms (LSVs) in Supplementary Fig. 2a. The prepared electrode exhibited the best catalytic activity with smallest Tafel slope

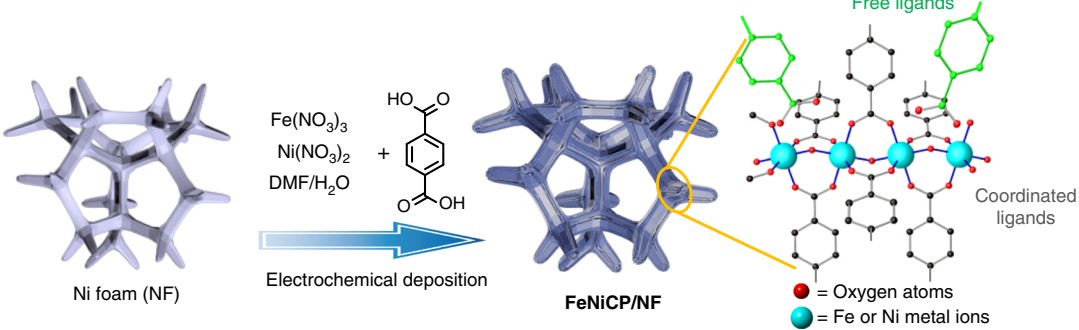

**Fig. 1** Schematic illustration of the fabrication procedure of the Ni-Fe coordination polymer prepared via an in situ electrochemical deposition method on Ni foam (NF) as a working electrode (**NiFeCP/NF**)

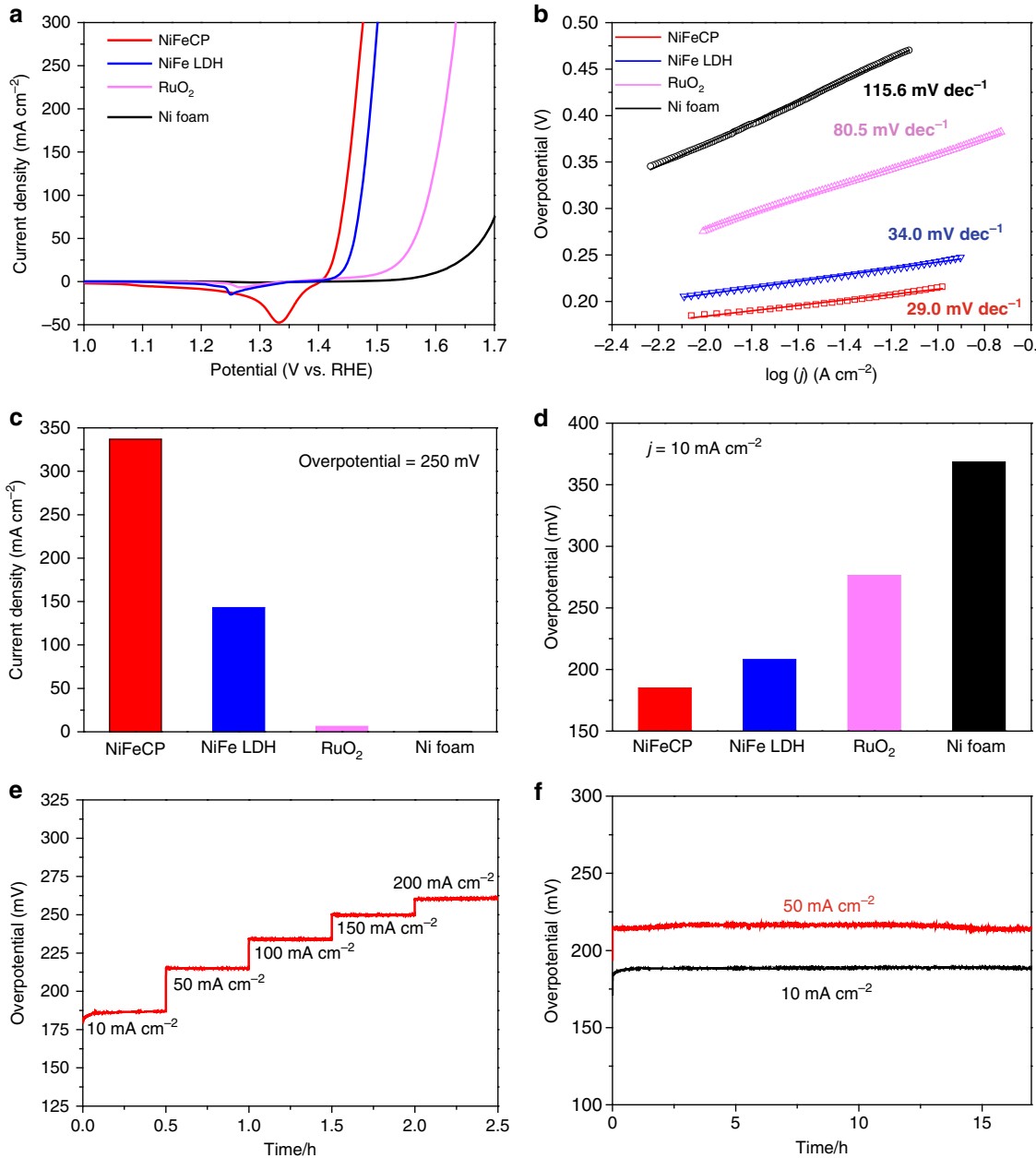

**Fig. 2** Electrochemical characterizations for OER. **a** Linear sweep voltammetry (LSV) curves of **NiFeCP/NF**, **NiFe LDH/NF**, RuO$_2$/NF, and Ni foam. **b** **NiFeCP/NF**, **NiFe LDH/NF**, RuO$_2$/NF, and Ni foam. **c** Current densities at an overpotential, $\eta = 250$ mV, as a function of electrode type. **d** The $\eta$ required for $j = 10$ mA cm$^{-2}$ as a function of electrode type. **e** Chronopotentiometric measurements of OER in 1.0 M KOH as a function of current density. **f** Extended chronopotentiometric measurements at $j = 10$ mA cm$^{-2}$ and $j = 50$ mA cm$^{-2}$ for 17 h

(Supplementary Fig. 2b) for OER when the Ni:Fe ratio was 7:3 in the electrodeposition solution, thus, this electrode was systematically studied in this work, and abbreviated as **NiFeCP/NF** in this paper.

For comparison, the state-of-the-art NiFe layered double hydroxide (**NiFe LDH**) catalyst was prepared on Ni foam (**NiFe LDH/NF**) as previously reported[21–23]. The polarization curves of **NiFeCP/NF**, **NiFe LDH/NF**, RuO$_2$/NF and nickel foam toward OER in 1.0 M KOH are shown in Fig. 2a. Among the electrodes, **NiFeCP/NF** electrode (the best samples) demonstrates an outstanding catalytic activity, delivers the highest current density at the same $\eta$ than the other electrodes.

In order to investigate the intrinsic activity of **NiFeCP/NF** and **NiFe LDH/NF** electrodes, the polarization curves are normalized to the electrochemically active surface area (ECSA), respectively

(Supplementary Fig. 3). **NiFeCP/NF** required the much lower overpotential for the same normalized current density, for example, **NiFeCP/NF** achieves a 10 mA cm$^{-2}$ normalized current density at $\eta = 190$, while **NiFe LDH/NF** required an overpotential of 210 mV. This result indicated that **NiFeCP/NF** displayed the better performance than that of **NiFe LDH/NF** electrode, which is because of that **NiFeCP** catalyst itself has a better intrinsic activity compared to that of **NiFe LDH**, rather than the larger specific surface area of **NiFeCP/NF** electrode.

Furthermore, **NiFeCP/NF** exhibits a remarkable low Tafel slope of 29 mV dec$^{-1}$, while **NiFe LDH/NF** shows a Tafel slope of 34 mV dec$^{-1}$ (Fig. 2b). The Tafel slope value for **NiFe LDH/NF** is in agreement with literature reports[21–23], indicating the reliability of our results. The various catalysts were further compared by a plot of their catalytic currents at a fixed $\eta$ of

250 mV. As shown in Fig. 2c, the **NiFeCP/NF** electrode delivers a current density of 337 mA cm$^{-2}$, which demonstrates as 2.3, 56, and 521−fold increase over that of **NiFe LDH/NF** (143 mA cm$^{-2}$), RuO$_2$/NF (6.2 mA cm$^{-2}$), and NF (0.64 mA cm$^{-2}$), respectively. To reach a current density of 10 mA cm$^{-2}$, the **NiFeCP/NF** electrode requires an overpotential of 188 mV, which is 22, 89, and 180 mV lower than that of **NiFe LDH/NF** (210 mV), RuO$_2$/NF (277 mV), and NF (368 mV), respectively (Fig. 2d).

To test steady-state activity and durability, the **NiFeCP/NF** electrode was subjected to a series of chronopotentiometry experiments comprising multiple current steps in 1.0 M KOH for 0.5 h. As shown in Fig. 2e, the corresponding required potentials are profiled when the catalytic current densities were increased from 10 to 200 mA cm$^{-2}$. At an initial current density of 10 mA cm$^{-2}$, an overpotential of 188 mV was needed, which remained constant for the duration of the test (0.5 h). Thereafter, the values of overpotential were observed to increase, and maintain their stability, when the current density was increased from 50 to 200 mA cm$^{-2}$. **NiFeCP/NF** displayed a current density of 50, 100, 150, and 200 mA cm$^{-2}$ at overpotentials of 214, 234, 249, and 260 mV, respectively.

Furthermore, the durability of the **NiFeCP/NF** electrode toward OER was examined at constant current densities of 10 and 50 mA cm$^{-2}$ for 17 h. As shown in Fig. 2f, overpotentials of 188 mV and 214 mV are needed to maintain the OER catalytic current densities at 10 mA cm$^{-2}$ and 50 mA cm$^{-2}$, respectively. The double-layer capacitance (C$_{dl}$) is a positive correlation with the ECSA of the **NiFeCP/NF** electrode, which was measured (Supplementary Fig. 4) after 17 hours electrolysis, no obvious change of C$_{dl}$ has be observed compared to the **NiFeCP/NF** electrode before electrolysis (Supplementary Fig. 5). The amount of O$_2$ generated by **NiFeCP/NF** and **NiFe LDH/NF** were measured, and Faradaic efficiencies of 98.4% and 97.8% were obtained at $j = 10$ mA cm$^{-2}$ for **NiFeCP/NF** and **NiFe LDH/NF**, respectively, indicating the accumulated charge passed through **NiFeCP/NF** and **NiFe LDH/NF** electrodes were almost quantitatively consumed for OER (Supplementary Fig. 6).

Transition metal coordination polymers such as metal–organic frameworks (Ni, Co, Fe MOFs) have been extensively studied as a new class of catalysts toward OER in alkaline solutions[24–31]. Unfortunately, only a few transition metal coordination polymers with low overpotential and excellent stability have been reported. Hitherto, Ni-Co bimetal organic framework nanosheets (NiCo−UMOFNs) electrode loaded on copper foam in the presence of a binder (Nafion solution) demonstrates a low onset potential of 1.39 V and an overpotential of 189 mV at 10 mA cm$^{-2}$ in alkaline conditions (according to LSV measurements)[31]. Ni-Fe bimetal two-dimensional (2D) ultrathin MOF nanosheets (NiFe−UMNs) have been reported as a catalyst for OER with 10 mA cm$^{-2}$ current density at an overpotential of 260 mV on a glass-carbon electrode[29]. NiFe MOF (MIL-53) nanosheets grown in situ on Ni foams via a solvothermal process, presented excellent activity toward OER with 50 mA cm$^{-2}$ current density at an $\eta$ of 233 mV[32]. Interestingly, the present **NiFeCP/NF** demonstrates enhanced performance when compared with MIL-53 nanosheets grown in situ on Ni foam via a solvothermal process (50 mA cm$^{-2}$ current density at an $\eta$ of 214 mV for **NiFeCP/NF**). Supplementary Table 1 lists the performance of **NiFeCP/NF** in the current work and previously reported electrodes for electrocatalytic water oxidation, including coordination polymer based and inorganic material based catalysts. This comparison demonstrates that the **NiFeCP/NF** electrode prepared via the fast electrochemical deposition process, is clearly superior to other reported systems derived from high crystallinity methods in terms of $\eta$ and the Tafel slope.

To elucidate the structural characteristics of **NiFeCP**, especially the existence of uncoordinated carboxyl groups, various techniques were performed to characterize the structure of as prepared **NiFeCP/NF** and after OER electrochemical experiments. The electrodeposition process appeared to generate a brown film deposited on the surface of the nickel foam, however, after OER, the color of the as prepared **NiFeCP/NF** electrode turned black (Supplementary Fig. 7), which could change back to a brown color after rinsing with ethanol or by placing it in air over a prolonged period of time. The color changing phenomenon is similar to that observed for **NiFe LDH**, indicating that the following reaction occurs: Ni-OH → Ni-OOH. Additionally, -Ni$^{II}$OH species could be present before and after OER in the presence of **NiFeCP/NF**. The X−ray powder diffraction (XRD) pattern (Supplementary Fig. 8) of the scratched **NiFeCP** powder from the electrode is characteristic of the reported MIL-53 diffractogram, and reveals that the as prepared **NiFeCP** film contains MOF components, where Ni atoms and Fe atoms are randomly arranged in the coordination polymer scaffold encompassed in large square-shaped pores[29,32,33]. However, other amorphous coordination polymer, metal-oxo species could not be excluded by the XRD diffractogram. After electrolysis, the MOF-like characteristic peaks associated with **NiFeCP** could not be observed in the XRD diffractogram, which may be a result of that excessive terephthalates in the as prepared **NiFeCP** film introduced by fast electrochemical deposition process can be removed during activation process.

The scanning electron microscopy (SEM) images (Supplementary Fig. 9), demonstrate that as prepared **NiFeCP/NF** is a 3D macroscopic film, which is uniformly covered on the surface of the **NF** skeleton. After the OER test, no obvious morphology change is observed for **NiFeCP/NF**.

Additionally, high-angle annular dark-field scanning transmission electron microscopy (HAADF-STEM) elemental mapping shows a homogeneous distribution of Ni, Fe, C, and O elements in pristine **NiFeCP** (Fig. 3a). After being subjected to OER electrochemical experiments, C, O, Ni, and Fe remain uniformly distributed in the post-OER **NiFeCP** (Fig. 3b). Energy-dispersive X-ray spectroscopy (EDS) shows that the Fe/Ni atomic ratio in **NiFeCP** is consistent at 1:9 both before and after OER (Supplementary Fig. 10); however, the component of carbon lost partially, hinting that the partial dissociation of the organic ligands during OER.

X-ray photoelectron spectroscopy (XPS) of C 1 s, O 1 s, Fe 2p, and Ni 2p before and after OER electrochemical experiments are presented in Supplementary Fig. 11a, Fig. 4. As shown in Fig. 4a, the C 1 s signal can be deconvoluted into three surface components, corresponding to the organic ligands at binding energies of 284.8 eV (the C–C bonds), 286.0 eV (the C–O bonds) and 288.8 eV (carboxylate O = C–O groups)[29,33]. The C 1 s binding energies are consistent before and after OER. After OER, the signal ascribed to K can be observed on the surface of **NiFeCP/NF**, which may derive from residual KOH from the electrolyte. As shown in Fig. 4b, the O 1 s signal can be deconvoluted by fitting different peaks at binding energies of around 529.5, 531.0, 531.6, 532.9, and 533.4 eV, which are attributed to oxygen atoms on metal-oxygen bonds, metal-hydroxyl species, carboxylate group of the organic ligands, and water, respectively[32–34]. The O 1 s binding energies are consistent before and after OER, however the relative intensity of metal-oxygen bonds and metal-hydroxyl species increased after OER. **NiFeCP** shows peaks at binding energies of 712 eV, and 725 eV, which are corresponded to Fe$^{3+}$ 2p$_{3/2}$ and 2p$_{1/2}$, respectively (Fig. 4c)[33]. **NiFeCP** shows characteristic peaks associated with the Ni$^{2+}$ oxidation state (Fig. 4d). Ni-O species associated with Ni atoms and the organic ligands are observed at binding energies of 855.5 eV (2p$_{3/2}$) and 873.0 eV

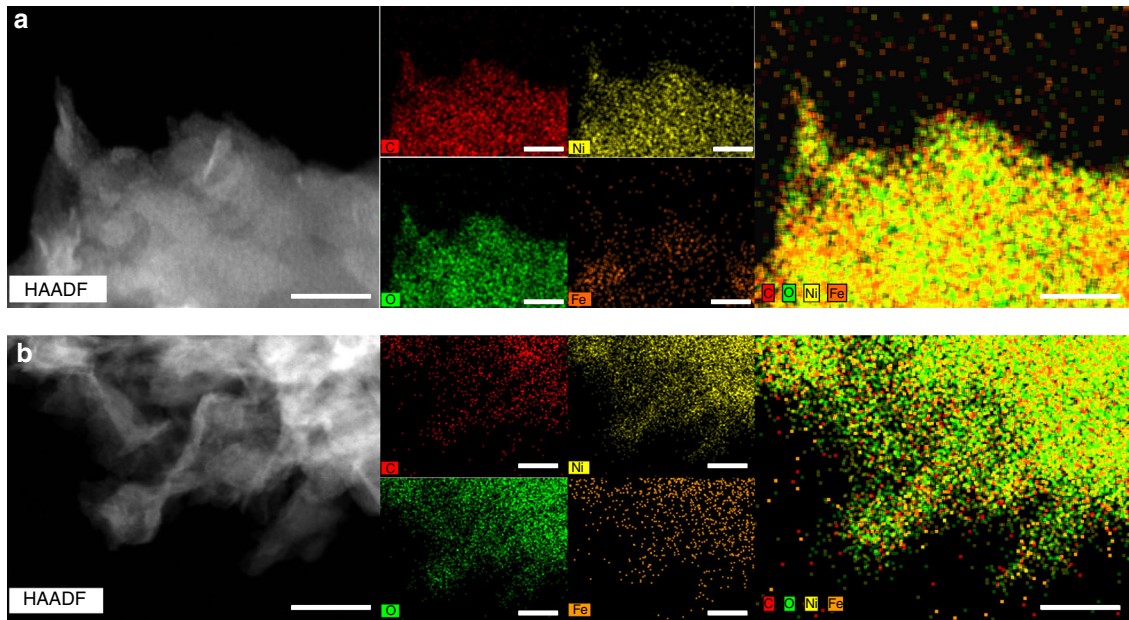

**Fig. 3** HAADF-STEM images and corresponding elemental mappings of **NiFeCP/NF**. Particles detached by sonication from **a** as prepared **NiFeCP/NF** electrode and **b** the electrode after OER test. The scale bars are 30 nm

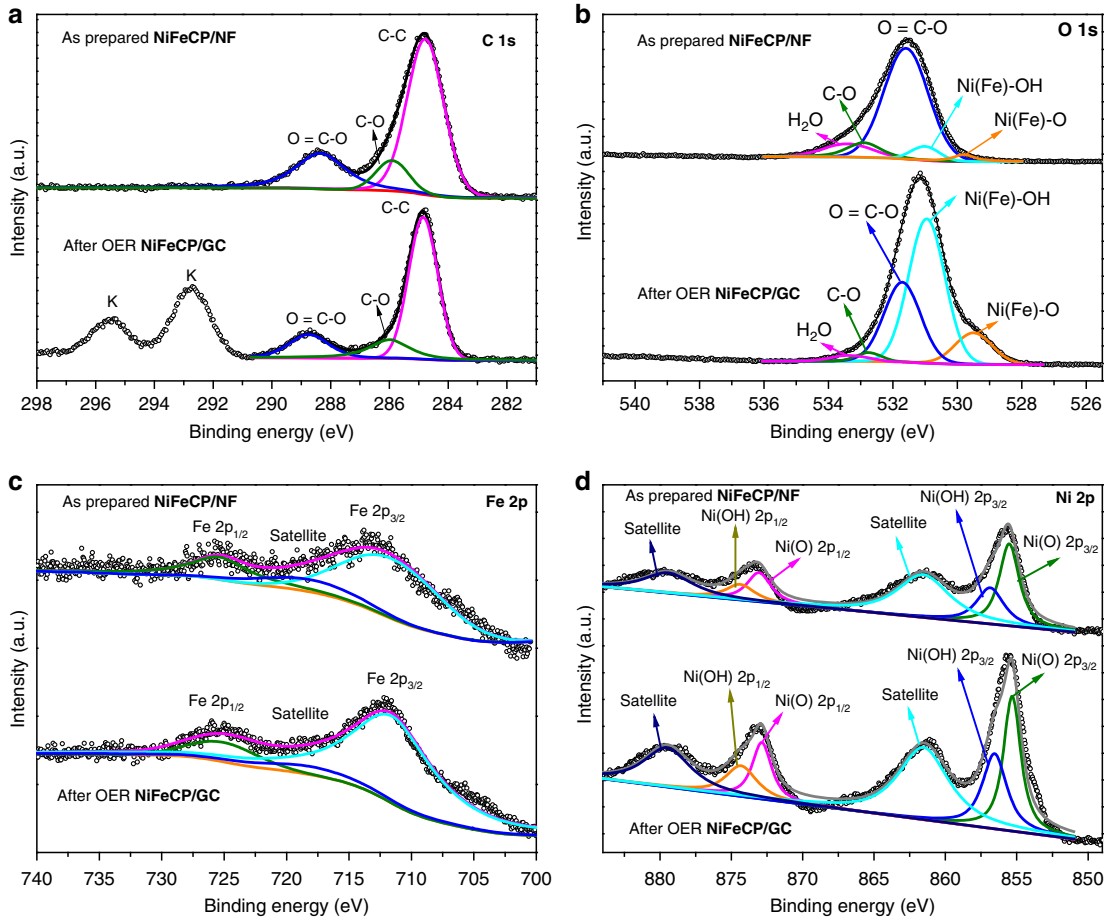

**Fig. 4** XPS measurements. High-resolution XPS spectra of **a** C 1 s, **b** O 1 s, **c** Fe 2p, and **d** Ni 2p for the **NiFeCP/NF** electrode before and after OER test

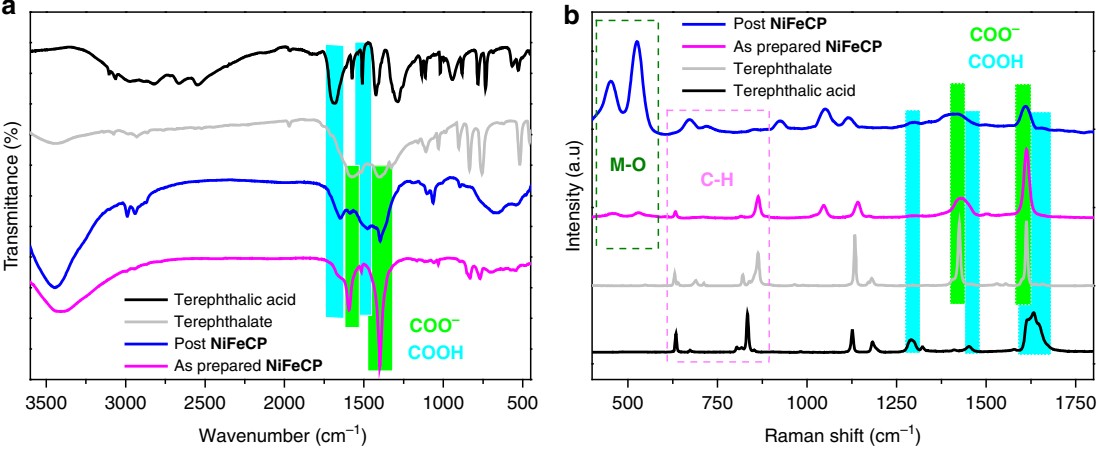

**Fig. 5** FT-IR and Micro-Raman measurements. **a** FT-IR spectra of dried **NiFeCP** powder scratched from the electrode before and after OER. **b** Micro-Raman spectra of **NiFeCP/NF** electrode before and after OER. Terephthalic acid and Na terephthalate as references

($2p_{1/2}$), while Ni-OH species are attributed to the binding energy peaks at 856.9 eV ($2p_{3/2}$) and 874.4 eV ($2p_{1/2}$)[32]. The presence of Ni-OH species proves the above-mentioned hypothesis regarding the change of color. The ratio of the integrated area associated with the Ni-OH/NiO peaks increased from 3:5 to 4:5 after electrolysis, and the ratio of metal-oxygen bonds and metal-hydroxyl species increased from O 1 s signal. In order to exclude the influences of Ni component form Ni foam, **NiFeCP** was also deposited on the surface of glassy carbon substrate (**NiFeCP@GC**) to study the surface oxidation states by XPS (Supplementary Fig. 11b). Without the influences of Ni foam, from the Ni 2p signal of high-resolution XPS spectra (Supplementary Fig. 12), the ratio for the relative intensity of metal-hydroxyl species and metal-oxygen bonds was found obviously increased after OER. The analysis of XPS spectra of the sample from **NiFeCP/NF** and **NiFeCP/GC** could give a consistent indication, that the carboxylate groups can partially de-coordinate from the metal centers of as prepared **NiFeCP**.

The Fourier transform infrared (FT-IR) spectra of powder scratched from as prepared **NiFeCP/NF** and the electrode after OER were investigated. As shown in Fig. 5a, two distinct peaks at 1384 cm$^{-1}$ and 1579 cm$^{-1}$, corresponding to the symmetric and asymmetric vibrations, respectively, of the carboxylate groups derived from the coordinated terephthalate, can be observed for the samples of as prepared and after OER[24]. Furthermore, two distinct absorption peaks at 1427 cm$^{-1}$ and 1684 cm$^{-1}$ are observed before and after OER corresponding to the uncoordinated carboxylate moiety of terephthalate[32]. The ratio of the uncoordinated carboxylate group increased after OER and is considered to have resulted by the de-coordination of the carboxylate groups from the metal centers. It is worth mentioning that for the MOF of MIL-53(FeNi) obtained by hydrothermal synthesis, the distinct absorption peaks of uncoordinated carboxylate can not be observed[32]. The Micro−Raman spectra (Fig. 5b) of **NiFeCP/NF** electrode exhibits a doublet at 1612 cm$^{-1}$ and 1429 cm$^{-1}$ before and after water oxidation, which corresponds to the in- and out- of phase stretching modes of the coordinated carboxylate groups, respectively. These two peaks of stretching modes were broad, which covered the moiety of uncoordinated carboxylate (1631 and 1451 cm$^{-1}$). Meanwhile, the vibration peak at 1293 cm$^{-1}$ ($A_g$ mode) of uncoordinated carboxylates can be obviously observed after OER[35], indicating the presence of uncoordinated carboxylates groups during OER. The intensity of the metal–O vibration peaks at 460 cm$^{-1}$ and 530 cm$^{-1}$ increase after electrolysis as a result of the de-

coordination of the carboxylate groups from the metal centers, and the generation of metal hydroxide species.

Combining the information derived from the above characterization suggests that part of the coordinated carboxylate groups can dissociated from **NiFeCP/NF**, and in doing so, forming additional free uncoordinated carboxylate sites and additional metal hydroxide sites during OER. Meanwhile, **NiFeCP/NF** contains metal hydroxide species, coordinated and uncoordinated carboxylate groups both before and after OER.

As the uncoordinated carboxylate group is always present in **NiFeCP** during water oxidation, we investigated the kinetic function of the uncoordinated carboxylates during water oxidation. As previously reported, increasing the electron density of the metal centers in a water oxidation catalyst may result in changes the Fermi level moving closer to the O 2p states, and concomitantly resulting in a transfer of the concerted proton-electron transfers process (c-PET) to the non-concerted proton-electron transfers(n-PET) process for OER[36].

As the carboxylate is a strong electron-donating ligand, it is necessary to determine the type of proton-coupled electron transfer processes for **NiFeCP** during water oxidation. Because for a catalyst to proceed via a n-PET pathway, the proton transfer process may not be involved in the rate determining step (RDS), therefore, the influence of the Lewis base will be difficult to be determined[37].

The study of the pH dependence on OER activity can provide useful insight on the kinetics and intermediates of the reaction. In this work, pH-dependence studies were performed in strongly basic solutions to probe the intrinsic activity of the catalysts[38]. The RHE scale was used to determine the pH dependence of the reaction kinetics to avoid the change in thermodynamic driving force at different pH values. The position of the redox peaks of **NiFeCP** and **NiFe LDH** are pH-dependent (Fig. 6a, b), as shown in Supplementary Fig. 13, the **NiFeCP** and **NiFe LDH** yielded linear plots of $E_{redox}$ (NHE) versus pH with slopes of −88 and −95 mV per pH, respectively. These values were nearly 1.5 times the theoretical value of −59 mV per pH for the 1 H$^+$/1e$^-$ oxidation of Ni$^{2+}$(OH)$_2$ to Ni$^{3+}$O(OH); thus, a 3 H$^+$/2e$^-$ coupled redox process was suggested for both **NiFeCP** and **NiFe LDH**[39–41]. The obtained slopes were in agreement with the previous report, whereby an Fe dopant could strongly decrease the valence state of the Ni in nickel hydroxide species[39,41]. For catalysts that proceed via a c-PET pathway, the reaction order of pH would be zero (see Supplementary Note 1 for the details), as the proton is never decoupled from the electron transfer in this

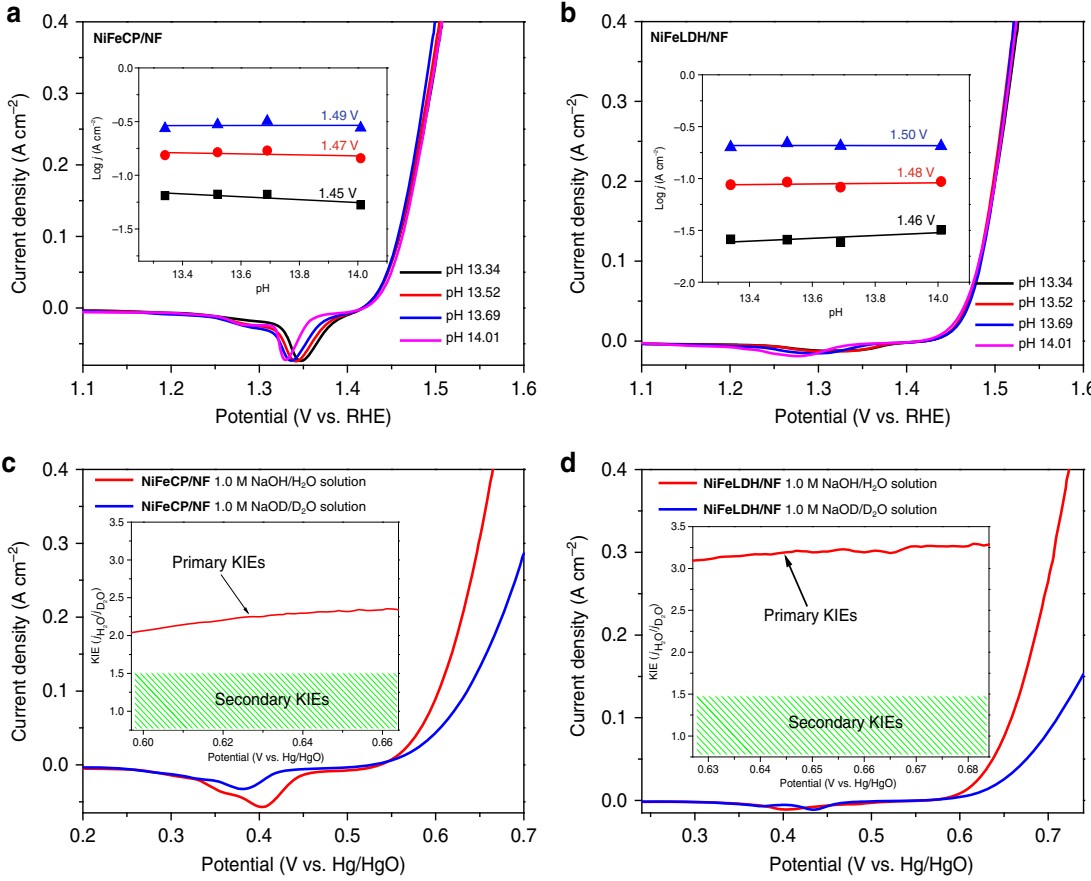

**Fig. 6** pH dependence and kinetic isotope effects studies. LSV curves of **a** NiFeCP/NF and **b** NiFe LDH/NF in KOH solutions as a function of pH. The inset exhibits the reaction order of pH value. LSV curves of: **c** NiFeCP/NF and **d** NiFe LDH/NF in aqueous 1.0 M NaOH solutions and 1.0 M NaOD D$_2$O solutions. The insets are the kinetic isotope effect values vs potential

case[36,38,42]. Both **NiFeCP** and **NiFe LDH** exhibit negligible pH-dependent OER kinetics (Fig. 6a, b), which indicates that both **NiFeCP** and **NiFe LDH** proceed via a c-PET process as the RDS during the water oxidation reaction, and naturally, the proton transfer must be involved[38].

Deuterium kinetic isotope effects (KIEs) can reflect the proton transfer kinetic information of water oxidation reactions, and therefore, help to interpret the RDS of the catalytic processes[43–45]. The presence of KIEs (KIEs > 1.5) is considered as evidence that proton transfer is involved in the RDS (or, at least, in one of the steps affecting the reaction rate)[46]. The KIEs experiments in 1.0 M NaOD D$_2$O solution were performed to obtain insight into the role of proton transfer during the catalytic RDS. The acidity difference between D$_2$O and H$_2$O caused by different dissociation constants was eliminated by the overpotential correction (see Supplementary Note 2 for an explanation). As the Faradaic Efficiency for both **NiFeCP/NF** and **NiFe LDH/NF** electrodes are close to 100% in aqueous and deuterated electrolytes (Supplementary Fig. 14), the accumulated charge for **NiFeCP/NF** and **NiFe LDH/NF** electrodes can be almost quantitatively consumed for OER, the corresponding current can be used directly for the calculation of KIEs. The LSV curve of **NiFeCP/NF** in a 1.0 M NaOD D$_2$O solution exhibits significantly lower current density in comparison with that of **NiFeCP/NF** in the 1.0 M NaOH H$_2$O solution by a factor about 2.2 over the entire potential range (Fig. 6c). The KIEs value of **NiFeCP/NF** indicates that the RDS for water oxidation involves cleavage of the O−H bonds. For the comparative sample, the **NiFe LDH/NF** shows a larger primary isotope effects (Fig. 6d). The pH-dependence and KIE studies

confirmed that the cleavage of the O−H bonds are involved for both **NiFeCP/NF** and **NiFe LDH/NF**. The proton vibrational wave function overlap plays an important role in determining the reaction rates and KIEs of c-PET reactions. Moreover, this overlap depends strongly on the proton donor−acceptor distance: the overlap is larger for shorter distances. Accordingly, the reaction rate increases and KIEs often decreases as the proton donor−acceptor distance decreases[47–49]. As both **NiFeCP/NF** and **NiFe LDH/NF** catalyze water oxidation by metal-oxo species with c-PET pathway, the smaller value of KIEs for **NiFeCP** than that of **NiFe LDH** suggests that there are functional groups nearby the catalytic centers to promote the kinetics of water oxidation. Further, anhydrous disodium terephthalate (0.3 M, almost saturated) was added into the electrolytes, the catalytic current of **NiFe LDH** slightly raised (~1.1 times) in 1.0 M NaOH H$_2$O solution (Supplementary Fig. 15a), obviously increased (~1.3 times) in 1.0 M NaOD D$_2$O solution (Supplementary Fig. 15b), resulting in the KIEs decreased from ~3.1 to 2.3 with the presence of terephthalate for the carboxylate free catalyst **NiFe LDH** (Supplementary Fig. 15c, d). When concentration of terephthalate in electrolytes is high, the carboxylates of terephthalate will have opportunity to help handling the proton transfer at the surface of **NiFe LDH**, for which, the carboxylates will serve the similar function as in the **NiFeCP** material, resulting in a smaller KIEs. This control experiment strongly proves the proton transfer relay promotion of the secondary coordination sphere effects caused by the uncoordinated carboxylates in **NiFeCP**.

To provide further support for the existence of proton transfer relays in **NiFeCP** for OER, electrochemical proton inventory

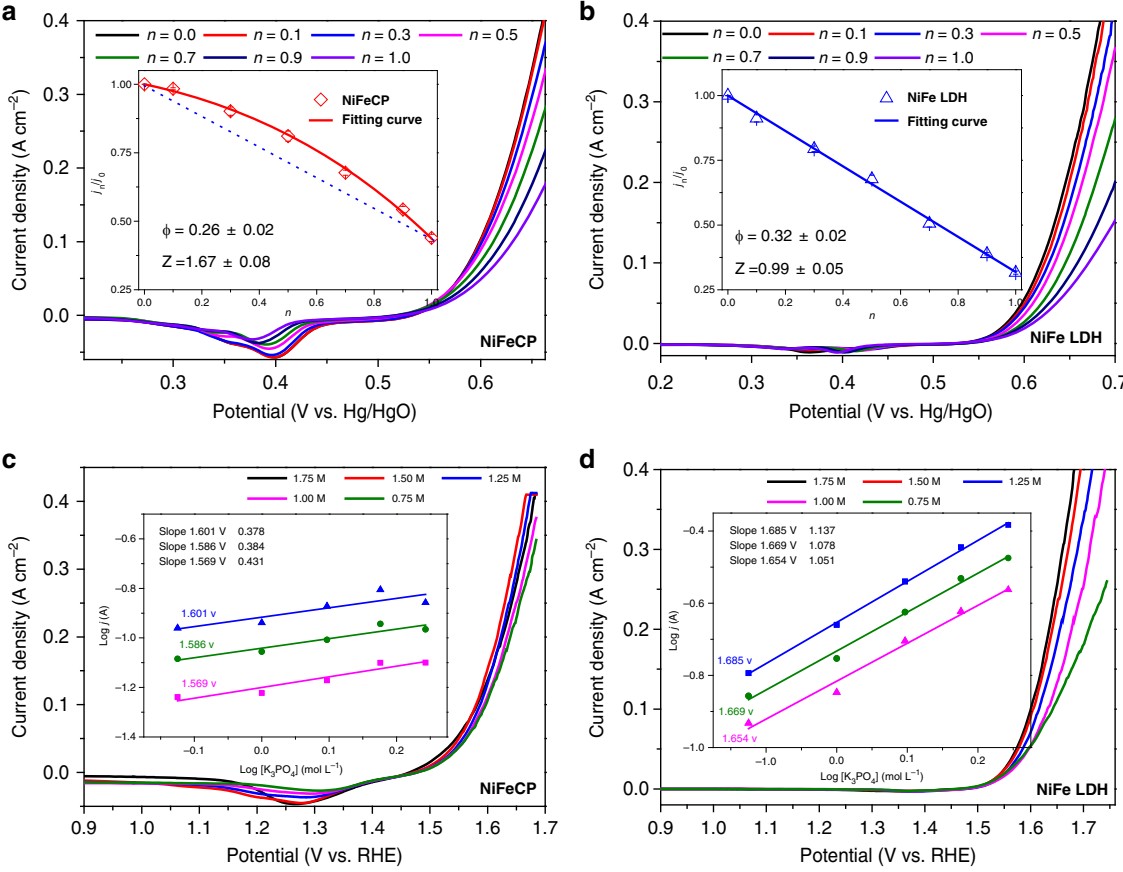

**Fig. 7** Proton inventory and atom proton transfer studies. LSV curves of **a NiFeCP/NF** and **b NiFe LDH/NF** in mixed solutions of 1.0 M NaOH in $H_2O$ and 1.0 M NaOD in $D_2O$ with different ratios as a function of atom fractions of deuterium (n). The insets exhibit the plots of $j_n/j_0$ as a function of n, where n = [D]/([D] + [H]) and at a certain potential were abbreviated as $j_n$, $j_0 = j_{H2O}$ (all reported values are averages from 12 different potentials, which can be find in Supplementary Note 3, all errors are given as standard deviations). LSV curves of: **c NiFeCP/NF** and **d NiFe LDH/NF** as a function concentration of $K_3PO_4$ at pH = 12.65, the insets exhibit the reaction order of $[K_3PO_4]$

studies were conducted. The proton inventory method is particularly useful for resolving the number of exchangeable hydrogenic sites that contribute to the catalytic rate[50–54]. The key parameter of proton inventory studies, isotope fractionation factor ($\phi$) is related to the propensity for a hydrogenic site in the RDS of the reaction to interact with $D^+$ compared to water, which only depends on the chemical structure in the immediate neighborhood of the isotopic site, and for a given functional group bearing exchangeable hydrogen will tend to have the same fractionation factors[53]. The dependence of the reaction rate attenuation on the atom fractions of deuterium in the electrolyte was measured (see Supplementary Note 3 for the details), and the data was fitted by a modified Kresge–Gross–Butler equation (Supplementary eqn. 8)[50–52], where $\phi$ is the isotopic fractionation factor and Z is the Z-effect related to the aggregate isotope effect from multiple equivalent hydrogenic sites, called Z-sites, with individual weak isotope effects[52,53]. In a plot of $j_n/j_0$ as a function of n, the shape of the resulting curve is dependent on the relative sizes of $\phi$ and Z. When Z > 1 suggests there is an aggregate inverse isotope effect at the Z-sites contributing to the observed kinetics, and Z ≈ 1 suggests there are no Z-sites contributing to the observed kinetics[51,52]. For **NiFeCP/NF** electrode (Fig. 7a), a nonlinear dome-shaped response with a $\phi$ of 0.26 and a Z of 1.6 can be observed. The large Z-effect indicated that the RDS of water oxidation at the active sites of **NiFeCP** coupled with an aggregate inverse-isotope effect from the Z-sites (uncoordinated carboxylate)[50–52]. However, as shown in Fig. 7b, the plot of $j_n/j_0$ as a function of n for the carboxylate-free **NiFe LDH/NF** electrode,

results in a linear attenuation with Z ≈ 1 and $\phi$ ≈ 0.32, which suggested that only one hydrogenic site (water) is involved in the RDS of the catalytic processes, and no proton relays contribute to the observed kinetics[52,55]. The smaller value of $\phi$ for **NiFeCP** indicated the uncoordinated carboxylate provided extra hydrogenic site, which influenced the transition-state hydrogen bridges corresponding to proton transfer in the RDS of OER, otherwise, a similar value of $\phi$ as **NiFe LDH** should be obtained[52,53]. Proton inventory studies strongly support the results obtained by KIEs measurements where the proton delivery from catalytic centers of **NiFeCP** to electrolyte received the assistance of uncoordinated carboxylates.

When the water oxidation RDS of a catalyst is related to the proton transfer, the atom proton transfer (APT), with a Lewis base in solution as a proton acceptor, can decrease the barrier of the reaction. Hence, Lewis bases (such as phosphate) in solution usually influence the reaction kinetics[56–58]. The relationship between catalytic activity and the concentration of additional base ($K_3PO_4$) were therefore studied. When an electrode undergoes a extra base-dependent pathway for water oxidation, because of the linear relationship between the water oxidation reaction rate $k_{cat}$ and catalytic current density, it should be first order reactions for the concentration of phosphate ($\rho_{phosphate}$) (see Supplementary Note 4 for an explanation)[59,60]. The first order reactions of phosphate have been widely observed in catalyst-modified electrodes for OER, when the RDS for water oxidation is clearly related to proton transfer[56,61]. As shown in Fig. 7d, the $\rho_{phosphate}$ for **NiFe LDH/NF** is around 1 over the entire obvious catalytic

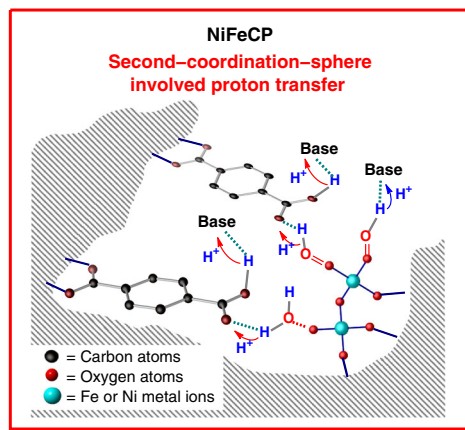
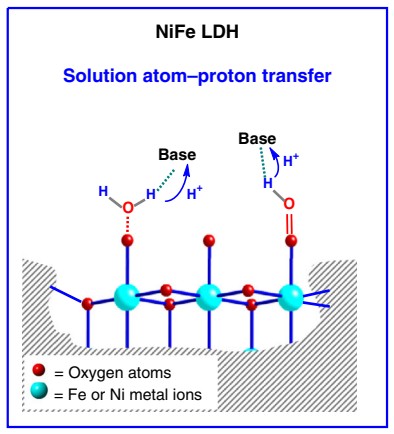

**Fig. 8** Schematic illustration of the proposed proton transfer processes: second-coordination-sphere involved proton transfer for **NiFeCP** (left) and APT for **NiFe LDH** (right)

potential range, indicating that solution APT can occur for the carboxylate-free **NiFe LDH/NF**. However, although the **NiFeCP/NF** is a catalyst that involves proton transfer event in the RDS, the $\rho_{phosphate}$ associated with **NiFeCP/NF** is significantly less than one (approximately 0.38) over the entire obvious catalytic potential range (Fig. 7c), which suggests the solution APT process attributed from the extra base is greatly suppressed. The solution APT process is a diffusion controlled event, and therefore, is influenced by the concentration of base, however, for **NiFeCP/NF**, the uncoordinated carboxylate groups are located in the vicinity of the catalytic center, and thus, fast proton exchange should preferentially occur, and thereafter, the external base from electrolyte can deprotonate the carboxylic acid produced from second−coordination−sphere involved proton transfer processes. Experimental results of **NiFeCP** catalyst have shown the smaller value of KIEs, non-linear response to the atom fraction of deuterium, and an effective suppression of solution APT in contrast to the carboxylate-free **NiFe LDH** catalyst, indicating that the uncoordinated carboxylate groups in **NiFeCP** provide internal bases, as shown in Fig. 8, which serve as a proton transfer relays.

## Discussion

The phosphate can serve to neutralize the protons generated by the c-PET electrode reaction and to convert, at least partially, proton diffusion toward the solution into a protonated phosphate diffusion toward the solution, and concomitantly, a diffusion of phosphate toward the electrode. Additionally, the phosphate can participate as a reactant to the c-PET electrode reaction[62]. Under buffer-free conditions, e.g., a 1.0 M KOH solution, APT-type catalysts have no buffer anion to assist proton transfer. However, for the catalysts, in which second-coordination-sphere is involved in proton transfer process, the internal base itself will provide the functions of a buffer anion to handle the proton, which can optimize the proton transfer during the reaction, and therefore increase reaction rate of water oxidation.

This second-coordination-sphere involved proton transfer pathway is certainly playing a key role during water oxidation in the OEC of PSII, as previously mentioned. The observation of the proton transfer relay phenomenon by uncoordinated carboxylate groups in the synthetic **NiFeCP** catalyst suggests that the second-coordination-sphere can be synthetically constructed in a heterogeneous artificial water oxidation catalyst, which is an interesting discovery. This may explain why **NiFeCP** and **NiFe LDH** are both water oxidation catalysts but proceed *via* different proton transfer pathways, and why the activity in terms of onset potential, overpotential and Tafel slope for **NiFeCP/NF** are

superior to those of **NiFe LDH/NF**. Certainly, further studies on second-coordination-sphere involving proton-coupled electron transfer catalysts are required to design and synthesis of more advanced heterogeneous water oxidation catalysts.

Inspired by the second-coordination-sphere of water oxidation catalyst in PSII, a coordination polymer containing Ni/Fe cations and terephthalate was electrochemically deposited in situ on a nickel foam as the functional electrode (**NiFeCP/NF**) for water oxidation. Various characterization techniques demonstrated that both coordinated and uncoordinated carboxylate groups in the film were maintained after electrolysis. **NiFeCP/NF** exhibits outstanding electrocatalytic water oxidation activity with a low overpotential of 188 mV at 10 mA cm$^{-2}$ in 1.0 M KOH, and a small Tafel slope of 29 mV dec$^{-1}$. To the best of our knowledge, the as-fabricated **NiFeCP/NF** electrode is one of the promising material catalysts for oxygen evolution. Comprehensive mechanism studies on **NiFeCP** and the benchmark OER catalyst of **NiFe LDH** were performed. **NiFeCP** and **NiFe LDH** exhibit pH-independent OER activities on the RHE scale, suggesting concerted proton-coupled electron transfer (c-PET) plays a great role in catalyzing the OER for both catalysts. Deuterium kinetic isotope effect measurements show a smaller value of KIEs for **NiFeCP** than that of carboxylate-free **NiFe LDH**; proton inventory studies show a non-linear dependence on deuterium concentration for **NiFeCP**; and APT measurements show a smaller external base impact from the electrolyte on **NiFeCP** than that of **NiFe LDH**. All experimental results suggest that the uncoordinated carboxylates can serve as proton transfer relays nearby the catalytic centers of **NiFeCP**. Such proton transfer relays can significantly improve the activity of the catalyst undergoing c-PET pathway for OER. This interesting discovery may provide a new perspective on the design and synthesis of more advanced heterogeneous catalysts for further enhancing catalytic activity through second-coordination-sphere engineering, and to form a base for next level research in the field of OER.

## Methods

**Preparation of NiFeCP/NF electrodes**. As a typical procedure, the electrochemical-deposition was carried out in a three-electrode electrochemical cell using a 1 × 2 cm$^2$ nickel foam as the working electrode, a platinum mesh (1 × 1.5 cm$^2$) as the counter electrode and a Hg/HgO as the reference electrode. Before electrodeposition, the nickel foam was cleaned in a HCl solution (1.0 M), rinsed with water and ethanol, then dried by N$_2$ flow. Preliminary electrocatalytic performances of **NiFeCP/NF** electrodes prepared by different Fe/Ni ratio in electrolyte were characterized, when Fe/Ni ratio equaled to 3/7 the **NiFeCP/NF** electrode displayed the best catalytic activity with the lowest overpotential about 180 mV for 10 mA cm$^{-2}$ and lowest Tafel slope of 29 mV dec$^{-1}$ for OER. In details, the electrochemical-deposition solution contained 0.6 mmol Terephthalic acid, 0.7 mmol Ni(NO$_3$)$_2$·6H$_2$O, 0.3 mmol Fe(NO$_3$)$_3$·9H$_2$O, 0.6 ml water and 10 ml DMF.

The **NiFeCP** films were prepared through a repeated double-current pulse chronopotentiometry (r-DCPC) method that involved two current pulses at $-3.0$ mA cm$^{-2}$ and 0 mA cm$^{-2}$ for 5 s and 10 s, respectively, as one DCPC deposition cycle. During this electro−deposition process, OH$^-$ was generated by the reduction of nitrate and water. With the increasing of the pH value nearby the working electrode, terephthalate acid can be deprotonated, and thereafter, react with metal cations forming a coordination polymer. High quality **NiFeCP** films can be obtained by repeating DCPC deposition after 600 cycles. After deposition, the **NiFeCP/NF** electrode was carefully withdrawn from the electrolyte, rinsed with DMF and ethanol, and then dried in air. Before water oxidation measurements, the as-prepared **NiFeCP/NF** electrodes were activated at 50 mA cm$^{-2}$ current density in 1.0 M KOH for 10 min to remove the excessive the terephthalates in **NiFeCP** introduced by the fast electrochemical deposition process. The activated **NiFeCP/NF** electrodes were rinsed with water, dried in air, and then measured the water oxidation activities in fresh electrolytes.

**Electrochemical Measurements**. Linear sweep voltammetry (LSV) measurements were carried out on a CHI 660e potentiostat. All the electrochemical characterizations were carried out at the temperature of 298 K, which was controlled by thermostatic water-bath. OER were studied in a standard three-electrode glass cell connected to CHI 660e workstation using the prepared materials as the working electrode, a Pt mesh as the counter electrode and a Hg/HgO electrode as the reference electrode. All the measured potentials were converted to reversible hydrogen electrodes (RHE) according to $E_{RHE} = E_{Hg/HgO} + 0.059$ pH $+ 0.0977$ V. The LSV measurements were performed at a scan rate of 5 mV s$^{-1}$ to evaluate the catalytic activities of the electrode films. All the polarization measurements were iR compensated (80%) unless otherwise stated. Chronopotentiometry measurements were recorded under the same experimental setup without iR correction.

**Determination of normalized LSV curves**. Electrochemically active surface area (ECSA) of electrode was obtained from cyclic voltammetry (CV) curves, In details, by plotting the $\Delta j$ ($|j_{charge} - j_{discharge}|$) at Faradaic silence potential range against the scan rates, the linear slope can be obtained, which is a positive correlation with the double-layer capacitance ($C_{dl}$), and been used to represent the corresponding ECSA.

**Determination of Faradaic Efficiency**. The faradaic efficiency was performed in a gastight electrochemical-cell using a Pt mesh as the counter electrode, **NiFeCP/NF** or **NiFe LDH/NF** as the working electrode and a Hg/HgO electrode as the reference electrode, respectively. 1.0 M KOH aqueous solution (or 1.0 M NaOD/D$_2$O solution) was used as electrolyte. Before the measurements, the electrolyte was purged with nitrogen gas for 30 min to completely remove the oxygen gas in the system. Then, the experiment was performed under the current density of 10 mA cm$^{-2}$. 0.5 mL of gas was analyzed by gas chromatography (GC, Techcomp GC 7890T, Ar carrier gas) after each 9.0 C of charge passed through the electrode.

## Data availability
Data underlying Figs. 2, 6 and 7. Supplementary Figs. 2–5 and 15 are provided as a Source Data file. All other data are available from the corresponding author upon reasonable request.

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

## Acknowledgements

This work was financially supported by the Fundamental Research Funds for the Central Universities (DUT19LK16), the Natural Science Foundation of China (Grant No. 21120102036, 91233201), the National Basic Research Program of China (973 program, 2014CB239402), the Swedish Research Council (2017-00935), the Swedish Energy Agency, and the K & A Wallenberg Foundation. We appreciate Dr. Ke Fan from Wuhan University of Technology for helpful suggestions. Open access funding provided by Royal Institute of Technology.

## Author contributions

F.L. and L.S. conceived the project, and wrote the manuscript. W.L. and F.L. performed the most of experiments and analyzed the date. H.Y., X.W., P.Z., and Y.S. helped in experiments and data analyses. All authors discussed the results and commented on the manuscript at all stages.

## Competing interests

The authors declare no competing interests.
