## [Peer Review File · Nature Communications]

Reviewers' comments:

Reviewer #1 (Remarks to the Author):

The authors report a NiFe coordination polymer material (NiFeCP) which when adsorbed on Ni foam electrodes shows promising activity and selectivity for the OER. The material oxidizes water at 187/188 mV overpotential at 10 mA/cm² current density and shows constant activity over 17 h of polarization. The authors characterized the material before-and-after OER tests with various techniques including XRD, HAADF-STEM, FTIR, TEM-EDS, and XPS. The authors compare this material to a NiFeLDH/NF material without a coordination polymer, and the NiFeCP/NF shows improved activity for OER. The authors show that NiFeLDH/NF has a KIE ~ 3 but NiFeCP/NF has a KIE ~ 2.1 . Both KIE suggest a rate-determining proton transfer step, but the smaller KIE for NiFeCP/NF was used to suggest there is a secondary-coordination sphere effect from uncoordinated terephthalate groups in the polymer that help OER at the NiFeCP/NF material as shown in Scheme 2.

This is a comprehensive study, and the inclusion of KIE studies is well considered. The activity of NiFeCP/NF for OER, while impressive, is not the main novelty of this paper—there are several materials deposited on Ni foam that show impressive comparable activity. Instead, the most interesting part of the manuscript was the postulated secondary coordination sphere effects that could be contributing to the enhanced activity of NiFeCP compared to NiFeLDH. Based on this alone, this manuscript has sufficient novelty and impact for publication in Nature Communications. However, the authors should address several comments below before publication.

1) The desire for an OER catalyst that operates with less than 200 mV overpotential at 10 mA/cm² seems to be an arbitrary distinction. Why do the authors feel that 200 mV overpotential is the important cut off for an active system? Second, the authors' catalyst operates at 10 mA/cm² at 189 mV. Many would argue that 187 mV is essentially the same as 200 mV...it is only 13 mV lower overpotential! The authors don't clarify in the main text if this is an average result or the "best" result, and they provide no standard deviations or any way for the readers to ascertain whether the 13 mV difference is statistically meaningful. Note that in the conclusions the authors report 188 mV overpotential, creating further ambiguity as to the standard deviation in the activity measurements. In addition, there are numerous NiFe-based systems on Ni foam that show comparable activity to the NiFeCP system (see for instance: Nat. Commun., 2015, 6, 6616.; J. Mater. Chem. A, 2016, 4, 13499-13508.; Chem. Mater., 2016, 28, 6934-6941; Small, 2018, 14, 1802204; J. Colloid Interface Sci., 2018, 523, 121-132; Yao et al., J. Power Sources, 2019, 424, 42-51; among dozens of others). I think the "outstanding electrocatalytic water oxidation activity" of the NiFeCP is the least important aspect of the paper, and should be put into better context by comparing to known reported systems. In addition, if one is to argue outstanding activity then one should make comparisons to other reported systems based also on mass activity or activity per surface area (see Chem Mater, 2017, 29, 120-140).

2) A smaller KIE for NiFeCP compared to NiFeLDH does not necessarily prove the existence of the secondary coordination sphere effect. Perhaps an important control experiment would be to conduct OER experiments with NiFeLDH and add terephthalic acid to the electrolyte and see whether the KIE decreases or remains constant (and also whether the activity increases or remains constant). If the KIE decreases and activity increases at high concentrations of terephthalic acid (where the mass-transport controlled local concentration would be similar to that in the NiFeCP material), this would support the idea that free carboxylic acid groups can increase catalytic activity by stabilizing reactive intermediates.

3) Electrochemical proton inventory studies may give additional information about the number of hydrogenic sites involved in the rate determining step (Inorg. Chem., 2017, 56, 11254-11265; Nat. Commun., 10, 2019, 1683). For example, according to Scheme 2 NiFeLDH should have one hydrogenic site (O-H in water) with an isotope fractionation factor ~ 0.33 (based on the KIE = 3).

However, in the NiFeCP sample there should be two hydrogenic sites (O-H in water and O-H in terephthalic acid), and thus should show a non-linear dependence on deuterium oxide concentration.

4) There is a peak at 1.1-1.32 V vs RHE in Figure 1a that is pH-dependent. What is this peak? The authors should discuss this peak and its pH-dependent shifts.

5) The authors do not provide convincing evidence that the MOF structure is maintained during OER. The XRD shows degradation to an amorphous structure post OER and the FTIR shows de-coordination of terephthalate from the metal centers. Both of these results are consistent with many other precursor films reported in the literature that degrade to amorphous materials upon oxidation. Given that the structure during and post OER is amorphous and unknown, the proposed proton-transfer processes shown in Scheme 2 may be misleading because they invoke a well-ordered structure.

6) Given that the authors support their catalyst on a high-surface area Ni foam, I would strongly caution the authors from interpreting their NiO and Ni-OH from XPS as belonging to the NiFeCP material. There is no indication that these Ni species are not from the Ni foam, and therefore attempting to derive information of the catalyst structure from these peaks or their ratios (e.g. de-coordination of carboxylate groups) may be an oversimplification of the system—although the NiFeCP is “scratched” from the Ni foam surface, there is no indication that some of the Ni foam itself was not scratched off as well. The authors may want to conduct an experiment using a planar, non-Ni based substrate (e.g. glassy carbon) to better quantify Ni oxidation states and ratios via XPS.

7) The authors should conduct electrolyses in D₂O to confirm the Faradaic Efficiency for O₂ production is the same as in the case of H₂O.

8) In Figure 5, the authors switch between units of mA/cm² and A/cm². Also, 5a, b, and d should probably be on the same y-axis scale, and 5c, e, and f should be on the same scale for easier comparison unless there is a specific reason otherwise.

9) The authors report the optimal Ni-Fe ratio for the NiFeCP/NF sample and use this in the OER studies. The authors should consider including the data for the other Ni-Fe ratios in the SI.

10) In Fig. S3, the y-axis label is not consistent with the figure caption. The label reads as oxygen concentration, but is described as Faradaic efficiency in the caption.

11) In the Raman spectra (Fig. 4b), the peaks for uncoordinated carboxylate groups at 1631 and 1451 wavenumbers don't seem to be present in the pristine and post NiFeCP samples, but in the main text the authors state that they are observed before and after the OER. The authors should clarify their statement to be consistent with Fig 4b.

Reviewer #2 (Remarks to the Author):

Comments on NCOMMS-19-14554 :

Recommendation: Published after minor revision.

Sun et al report “A Bio-inspired Coordination Polymer, NiFeCP/NF, as Outstanding Water Oxidation Catalyst via Second Coordination Sphere Engineering”. This catalytic system exhibits a low overpotential of 187 mV at 10 mA cm⁻² in 1.0 KOH, as well as a small Tafel slope and excellent stability. Interestingly, this excellent activity is related to the second sphere uncoordinated carboxylates which is serving as the proton transfer relays during the catalytic process. The design of this work is novel and give a simple but effective way to construct new material catalyst for water

splitting. In addition, the second sphere uncoordinated carboxylates play the role as similar to the amino acid in PSII, this work also gives a new way to mimic the function of amino acid in the PSII. The catalyst used in this work was well characterized and the electrochemical performance was also well investigated. According to the comparison of different reported WOCs, this bio-inspired Coordination Polymer, NiFeCP/NF, gives an impressive catalytic performance. This is an elegant work in this field and I suggest this work to be accepted for publication on Nature Comm.

Minor points:

1. If added the extra carboxylate into the catalytic system, what is the relationship between water oxidation reaction rate k_{cat} and the concentration of [carboxylate] if the uncoordinated carboxylate is the proton relay?
2. Is it possible to tune the catalytic activity by tuning the pK_a of uncoordinated carboxylate?

Reviewer #3 (Remarks to the Author):

The paper of Li et al. reports on an iron/nickel terephthalate coordination polymer deposited on nickel foam (NiFeCP/NF) to catalyze the anodic oxygen evolution reaction in alkaline media. Proposed in this work catalyst requires an overpotential of 188 mV at 10 mA cm^{-2} and exhibits a Tafel slope of 29 mV dec $^{-1}$ and, thus, shows better performance compared to the chosen benchmark material NiFe LDH. However, NiFe-based catalysts with even lower Tafel slopes were reported in literature (e.g. catalyst with Tafel slope of 15 mV dec $^{-1}$ in J. Phys. Chem. C 2008, 112, 3655-3666). Overall approach of incorporation of a catalyst in a polymer matrix can be interesting, but there are concerns regarding the stability of such polymer on the long run under the OER conditions. Polymers are typically unstable in the alkaline media, and their stability is drastically decreasing at high anodic potentials because the functional groups providing conductivity are prone to be attacked by the OH radicals. This challenge is very difficult to overcome and this is hindering development of membrane for alkaline water electrolysis in particular. There is no evidence in the current manuscript that the polymer matrix is stable during long term operation. Moreover, the HAADF-STEM images presented in Figure 2 suggest decrease in Carbon content after the electrolysis which indicates the degradation of the polymer. The data on electrochemical stability of the polymer itself is crucial, still it is not provided in the manuscript. The main question is what would be the gain in incorporating the catalyst into potentially unstable polymer matrix if already reported in the literature NiFe-based catalysts have comparable or even better performance? I doubt that without this important information the manuscript can influence the thinking in the field of electrocatalysis, as one would expect from the paper published in Nature Communications.

Another important point that is missing in the manuscript is comparison of the electrochemically active surface area before the anodic scan and after it. From the XRD data presented in Figure S5, it becomes clear that the OER leads to amorphization of the catalytic surface, which will affect the ECSA. Considering that polarization curves presented in Fig. 2 are taken till relatively high current densities the amorphization is probably ongoing while the polarization curve is being recorded and related to it increase of ECSA may significantly affect the measured reactivity.

- The XPS fitting model does not really match the experimental curve and some of important features are not considered in Figure 3. It is well known that anodic polarization of Ni-Fe based materials leads to formation of oxyhydroxides. However, the peak corresponding to the OH groups is missing in the XPS fitting model of O 1s level presented in Figure 3b. This should be corrected, since hydroxy species are included in the spectra of Ni and Fe in Figure 3. Also the fitting model doesn't include contribution of several important features, e.g. fitted line doesn't include small shoulder at ca 529 eV in Figure 3b after OER. The additional component at 583 eV is missing in the fitting model of Ni spectrum.

Minor questions

- what was the collection efficiency in GC measurements to estimate the efficiency of the OER? Ni and Fe are both stable towards dissolution under anodic polarization in the alkaline medium. What electrochemical process is responsible for the rest 4% of the current? Polymer oxidation/degradation?

Reviewer 1.

Reviewer's Comments:

The authors report a NiFe coordination polymer material (NiFeCP) which when adsorbed on Ni foam electrodes shows promising activity and selectivity for the OER. The material oxidizes water at 187/188 mV overpotential at 10 mA/cm² current density and shows constant activity over 17 h of polarization. The authors characterized the material before-and-after OER tests with various techniques including XRD, HAADF-STEM, FTIR, TEM-EDS, and XPS. The authors compare this material to a NiFeLDH/NF material without a coordination polymer, and the NiFeCP/NF shows improved activity for OER. The authors show that NiFeLDH/NF has a KIE ~ 3 but NiFeCP/NF has a KIE ~2.1. Both KIE suggest a rate-determining proton transfer step, but the smaller KIE for NiFeCP/NF was used to suggest there is a secondary-coordination sphere effect from uncoordinated terephthalate groups in the polymer that help OER at the NiFeCP/NF material as shown in Scheme 2.

This is a comprehensive study, and the inclusion of KIE studies is well considered. The activity of NiFeCP/NF for OER, while impressive, is not the main novelty of this paper—there are several materials deposited on Ni foam that show impressive comparable activity. Instead, the most interesting part of the manuscript was the postulated secondary coordination sphere effects that could be contributing to the enhanced activity of NiFeCP compared to NiFeLDH. Based on this alone, this manuscript has sufficient novelty and impact for publication in Nature Communications. However, the authors should address several comments below before publication.

Response: First, we sincerely thank this referee for your time, effort, and insights on our manuscript. In particular, the constructive comments and suggestions below are very helpful for us to revise and improve our manuscript. We have studied your comments carefully and performed some additional experiments accordingly to address your concerns.

Question 1:

The desire for an OER catalyst that operates with less than 200 mV overpotential at 10 mA/cm² seems to be an arbitrary distinction. Why do the authors feel that 200 mV overpotential is the important cut off for an active system? Second, the authors' catalyst operates at 10 mA/cm² at 189 mV. Many would argue that 187 mV is essentially the same as 200 mV. It is only 13 mV lower overpotential! The authors don't clarify in the main text if this is an average result or the "best" result, and they provide no standard deviations or

any way for the readers to ascertain whether the 13 mV difference is statistically meaningful. Note that in the conclusions the authors report 188 mV overpotential, creating further ambiguity as to the standard deviation in the activity measurements. In addition, there are numerous NiFe-based systems on Ni foam that show comparable activity to the NiFeCP system (see for instance: Nat. Commun., 2015, 6, 6616.; J. Mater. Chem. A, 2016, 4, 13499-13508.; Chem. Mater., 2016, 28, 6934-6941; Small, 2018, 14, 1802204; J. Colloid Interface Sci., 2018, 523, 121-132; Yao et al., J. Power Sources, 2019, 424, 42-51; among dozens of others). I think the “outstanding electrocatalytic water oxidation activity” of the NiFeCP is the least important aspect of the paper, and should be put into better context by comparing to known reported systems. In addition, if one is to argue outstanding activity then one should make comparisons to other reported systems based also on mass activity or activity per surface area (see Chem Mater, 2017, 29, 120-140).

Response: We fully agree with this referee that “the most interesting part of the manuscript was the postulated secondary coordination sphere effects that could be contributing to the enhanced activity of NiFeCP compared to NiFeLDH”; and “outstanding electrocatalytic water oxidation activity of the NiFeCP is the least important aspect of the paper.” Comparatively, our description in the original manuscript was suspicious for overselling the activity of NiFeCP/NF for OER. About the “200 mV overpotential is the important cut off”, we have changed the description to make the statements more clear, all changes in the main text are marked in color. We have added more discussions in “Introduction” section to explain why a comprehensive understanding of the underlying mechanism of proton-coupled interfacial electron transfer process in the rate-determining step is vitally important. We have also clarified in the main text about the “best result” of our catalyst.

Question 2:

A smaller KIE for NiFeCP compared to NiFeLDH does not necessarily prove the existence of the secondary coordination sphere effect. Perhaps an important control experiment would be to conduct OER experiments with NiFeLDH and add terephthalic acid to the electrolyte and see whether the KIE decreases or remains constant (and also whether the activity increases or remains constant). If the KIE decreases and activity increases at high concentrations of terephthalic acid (where the mass-transport controlled local concentration would be similar to that in the NiFeCP material), this would support the idea that free carboxylic acid groups can increase catalytic activity by stabilizing reactive intermediates.

Response:

Figure S15. (a) LSV curves of **NiFe LDH/NF** in aqueous 1.0 M NaOH/H₂O solution and 1.0 M NaOH/H₂O solution with 0.3 M anhydrous disodium terephthalate. The inset is the contradistinction of current density vs potential. (b) LSV curves of **NiFe LDH/NF** in aqueous 1.0 M NaOD/D₂O solution and 1.0 M NaOD/D₂O solution with 0.3 M anhydrous disodium terephthalate. The inset is the contradistinction of current density vs potential. (c) LSV curves of **NiFe LDH/NF** in aqueous 1.0 M NaOH/H₂O solution with 0.3 M anhydrous disodium terephthalate and 1.0 M NaOD/D₂O with 0.3 M solution with anhydrous disodium terephthalate. The inset is the KIEs values vs potential. (d) The KIEs vs potential of **NiFe LDH/NF** with and without terephthalate.

We have performed such control OER experiments according to the referee's suggestions, and positive results have been obtained as shown in Figure S15. Related discussions are made in the revised manuscript on page 15.

Question 3:

Electrochemical proton inventory studies may give additional information about the number of hydrogenic sites involved in the rate determining step (Inorg. Chem., 2017, 56, 11254-11265; Nat. Commun., 10, 2019, 1683). For example, according to Scheme 2 NiFeLDH should have one hydrogenic site (O-H in water) with an isotope fractionation factor ~ 0.33 (based on the KIE = 3). However, in the NiFeCP sample there should be two hydrogenic sites (O-H in water and O-H in terephthalic acid), and thus should show a non-linear dependence on deuterium oxide concentration.

Response: According to the constructive suggestions and the relevant literatures provided, we have carefully performed an additional experiment about the proton inventory studies, and the results are shown in Figure 6.

Figure 6. LSV curves of a) NiFeCP/NF and b) NiFe LDH/NF in mixed solutions of 1.0 M NaOH in H₂O and 1.0 M NaOD in D₂O with different ratios as a function of atom fractions of deuterium (n). The inset exhibits the plots of j_n/j_0 as a function of n , where $n = [D]/([D]+[H])$ and at a certain overpotential of η were abbreviated as $j_n, j_0 = j_{H_2O}$.

NiFeLDH showed a linear dependence on the atom fractions of deuterium, where the isotope fractionation factor $\Phi \approx 0.33$ and the Z-sites effect $Z \approx 1$, suggests there are no Z-sites contributing to the observed kinetics. As predicted by the referee, NiFeCP showed a non-linear dependence on the atom fractions of deuterium, where $Z > 1$, indicating an internal hydrogenic site involved in the rate determining step. By combining this proton inventory studies experiment with the pH-independence OER activities, KIEs and atom proton transfer measurements, solid evidences can be provided now to prove that the uncoordinated carboxylate can serve as proton relay in NiFeCP. Related results and discussions have been added into the revised manuscript on page 16.

Question 4:

There is a peak at 1.1-1.32 V vs RHE in Figure 1a that is pH-dependent. What is this peak? The authors should discuss this peak and its pH-dependent shifts.

Response: The position of the redox peak in Figure 1a is indeed pH-dependent. As illustrated in Figures 5a and 5b, the NiFeCP and NiFeLDH yielded linear plots of E_{redox} versus pH with slopes of -93 and -92 mV per pH, respectively. These values were nearly 1.5 times the theoretical value of -59 mV per pH for the $1H^+/1e^-$ oxidation of $Ni^{2+}(OH)_2$ to $Ni^{3+}O(OH)$; thus, a $3H^+/2e^-$ coupled redox process was suggested for both NiFeCP and NiFeLDH. The obtained slopes were in agreement with the previous reports, whereby an Fe dopant could strongly decrease the valence state of the Ni in Nickel hydroxide species. (*Proc. Natl. Acad. Sci. U.S.A.* 2010, 107, 10337. *J. Am. Chem. Soc.* 2013, 135, 12329.

Figure S13. Corresponding position of redox peaks (vs normal hydrogen electrode, NHE) extracted from LSV curves (**Figure 5a** and **5b** in the main text). (a) for NiFeCP/NF and (b) for NiFeLDH/NF.

The related discussions have been added to the revised manuscript.

Question 5:

The authors do not provide convincing evidence that the MOF structure is maintained during OER. The XRD shows degradation to an amorphous structure post OER and the FTIR shows de-coordination of terephthalate from the metal centers. Both of these results are consistent with many other precursor films reported in the literature that degrade to amorphous materials upon oxidation. Given that the structure during and post OER is amorphous and unknown, the proposed proton-transfer processes shown in Scheme 2 may be misleading because they invoke a well-ordered structure.

Response: We fully agree with this referee. The schematic structure in Scheme 2 has been revised into non-ordered structure. We hope such modified illustration could avoid any misleading.

Question 6:

Given that the authors support their catalyst on a high-surface area Ni foam, I would strongly caution the authors from interpreting their NiO and Ni-OH from XPS as belonging to the NiFeCP material. There is no indication that these Ni species are not from the Ni foam, and therefore attempting to derive information of the catalyst structure from these peaks or their ratios (e.g. de-coordination of carboxylate groups) may be an oversimplification of the system—although the NiFeCP is “scratched” from the Ni foam surface, there is no indication that some of the Ni foam itself was not scratched off as well. The authors may want to conduct an experiment using a planar, non-Ni based substrate (e.g. glassy carbon) to better quantify Ni oxidation states and ratios via XPS.

Response: According to the suggestions of this referee, NiFeCP on glassy carbon substrate (**NiFeCP@GC**) has been prepared, and the Ni oxidation states were measured by XPS. Without the influences of Ni foam, the ratio for the relative intensity of metal-hydroxyl species and metal-oxygen bonds was found obviously increased after OER. The ratio of the integrated area associated with the Ni-OH/NiO peaks increased from 6:10 to 7.2:10 after electrolysis for the **NiFeCP@GC** sample, which is similar to the **NiFeCP@NF** sample.

Figure S12. High-resolution XPS spectra of (a) C 1s, (b) O 1s, (c) Fe 2p, and (d) Ni 2p for particles

detached by sonication from the as prepared NiFeCP/GC and NiFeCP/GC after 5 h OER test.

Question 7:

The authors should conduct electrolyses in D₂O to confirm the Faradaic Efficiency for O₂ production is the same as in the case of H₂O.

Response: The Faradaic Efficiency for O₂ production in D₂O has been measured by electrolysis, the result is the same as in the case of H₂O, see Figure S14.

Figure S14. The Faradaic efficiency of (a) NiFeCP/NF and (b) NiFe LDH/NF for OER in a 1.0 M NaOD D₂O solution. Comparison of the amount O₂ of the theoretically calculated and experimentally measured gas versus quantity of electric charge for water splitting catalyzed by the NiFeCP/NF and NiFe LDH/NF at $j = 10 \text{ mA cm}^{-2}$.

Question 8:

In Figure 5, the authors switch between units of mA/cm² and A/cm². Also, 5a, b, and d should probably be on the same y-axis scale, and 5c, e, and f should be on the same scale for easier comparison unless there is a specific reason otherwise.

Response: Units and scales of figures have been unified in the manuscript.

Question 9:

The authors report the optimal Ni-Fe ratio for the NiFeCP/NF sample and use this in the OER studies. The authors should consider including the data for the other Ni-Fe ratios in the SI.

Response: Data for the other Ni-Fe ratios have been added in SI as shown in Figure S2.

Figure S2. a) Polarization curves of **NiFeCP/NF** with different proportions of Ni:Fe in the electrodeposition solution for OER, measured in 1.0 M KOH solution at a scan rate of 5 mV s⁻¹. b) Tafel plots for **NiFeCP/NF** with different proportions of Ni:Fe in the electrodeposition solution for OER, calculated from the data of **Figure S2a**.

Question 10:

In Fig. S3, the y-axis label is not consistent with the figure caption. The label reads as oxygen concentration, but is described as Faradaic efficiency in the caption.

Response: This error has been corrected in the revised manuscript.

Question 11:

In the Raman spectra (Fig. 4b), the peaks for uncoordinated carboxylate groups at 1631 and 1451 wavenumbers don't seem to be present in the pristine and post NiFeCP samples, but in the main text the authors state that they are observed before and after the OER. The authors should clarify their statement to be consistent with Fig 4b.

Figure 4 (b) Micro-Raman spectra of as prepared NiFeCP/NF and after OER electrochemical

experiments with terephthalic acid and Na terephthalate as references.

Response: The Micro-Raman spectra (Fig. 4b) of NiFeCP/NF exhibits a doublet at 1612 cm^{-1} and 1429 cm^{-1} before and after water oxidation, which correspond to the in- and out-of phase stretching modes of the coordinated carboxylate groups, respectively. These two peaks of stretching modes were broad, and covered the moiety of uncoordinated carboxylate (1631 and 1451 cm^{-1}). Meanwhile, the vibration peak at 1293 cm^{-1} (A_g mode) of uncoordinated carboxylates can be obviously observed after OER (*J. Mol. Struct.* 1997, 415, 93.), indicating that the presence of uncoordinated carboxylates groups during OER. Related statements have been corrected in the revised manuscript.

Reviewer 2.

Reviewer's Comments:

Recommendation: Published after minor revision.

Sun et al report "A Bio-inspired Coordination Polymer, NiFeCP/NF, as Outstanding Water Oxidation Catalyst via Second Coordination Sphere Engineering". This catalytic system exhibits a low overpotential of 187 mV at 10 mA cm⁻² in 1.0 KOH, as well as a small Tafel slope and excellent stability. Interestingly, this excellent activity is related to the second sphere uncoordinated carboxylates which is serving as the proton transfer relays during the catalytic process. The design of this work is novel and give a simple but effective way to construct new material catalyst for water splitting. In addition, the second sphere uncoordinated carboxylates play the role as similar to the amino acid in PSII, this work also gives a new way to mimic the function of amino acid in the PSII.

The catalyst used in this work was well characterized and the electrochemical performance was also well investigated. According to the comparison of different reported WOCs, this bio-inspired Coordination Polymer, NiFeCP/NF, gives an impressive catalytic performance. This is an elegant work in this field and I suggest this work to be accepted for publication on Nature Comm.

Response to comments:

First of all, we sincerely appreciate your very positive comments on our work. Your constructive suggestions are very helpful for us to improve the quality of this work and we have revise our manuscript accordingly.

Question 1:

If added the extra carboxylate into the catalytic system, what is the relationship between water oxidation reaction rate k_{cat} and the concentration of [carboxylate] if the uncoordinated carboxylate is the proton relay?

Response: First, we believe that there should be a best ratio between the free-carboxylate and the metal catalytic centers. Suitable amount of proton relay can accelerate the proton transfer rate, however, if too much free-carboxylates are added into the catalyst composites, it will reduce the ratio of active sites. Thence, water oxidation reaction rate of metal/carboxylate composites and the concentration of [carboxylate] in the composites should be a volcano-like relationship. Because of bimetallic Fe-Ni composites were used in our work, the amount of terephthalate will not only influence the concentration of carboxylate in NiFeCP, but also the ratio of Ni/Fe, which will make the

comparison of activities very complicated. Meanwhile, we have tried to increase the amount of terephthalate in NiFeCP according to your suggestion, however, because of the solubility of terephthalate in the electrodeposition solutions is low, it is difficult to added more carboxylate into NiFeCP.

Second, according to your inspired suggestion, extra experiments have been conducted. For the carboxylate free catalyst NiFeLDH, when 0.3 M anhydrous disodium terephthalate (almost saturated in 1.0 M NaOH) was added into the electrolyte, the catalytic current of NiFeLDH was raised by ca.1.1 times at range of obviously OER potentials. However, for NiFeLDH, when terephthalate was added to the electrolyte, the KIEs decreased. The KIEs decrease and activity increase at high concentration of terephthalate, this experiment supports the idea that free carboxylic acid groups can increase catalytic activity by serving as proton relay, which could accelerate the reaction rate of water oxidation by internal atom proton transfer mechanism.

Figure S15. (a) LSV curves of NiFe LDH/NF in aqueous 1.0 M NaOH/H₂O solution and 1.0 M NaOH/H₂O solution with 0.3 M anhydrous disodium terephthalate. The inset is the contradistinction of current density vs potential. (b) LSV curves of NiFe LDH/NF in aqueous 1.0 M NaOD/D₂O solution and 1.0 M NaOD/D₂O solution with 0.3 M anhydrous disodium terephthalate. The inset is the contradistinction of current density vs potential. (c) LSV curves of NiFe LDH/NF in aqueous 1.0 M NaOH/H₂O solution with 0.3 M anhydrous disodium terephthalate and 1.0 M NaOD/D₂O with 0.3 M solution with anhydrous disodium terephthalate. The inset is the KIEs values vs potential. (d) The KIEs vs potential of NiFe LDH/NF with and without with terephthalate.

Summarizing the pH-independence OER activities, KIEs, proton inventory studies and atom proton transfer measurements, solid evidences have been provided now to prove that the uncoordinated carboxylate can serve as proton relay in NiFeCP.

Question 2:

Is it possible to tune the catalytic activity by tuning the pKa of uncoordinated carboxylate?

Response: In PSII, the pKa of the bound water changes drastically resulting in deprotonation and formation of a bound hydroxide ($\text{Mn-OH}_2 \rightarrow \text{Mn-OH} + \text{H}^+$). The net effect of this hypothetical sequence of events would be hydroxide binding to the OEC. The electric field emanating from the positive charge at Yz^+ (Tyr160/161) affects pKa values of bound water and drives several proton hopping steps and eventually results in a transfer of a proton from the OEC to the aqueous phase. Oxidation of the OEC directly coupled to a proton transfer to a water molecule is an unlikely event in the acidic and neutral pH regime because a water molecule is an unfavorable proton acceptor (Biochim. Biophys. Acta Bioenerg. 2007, 1767, 472; ChemCatChem 2010, 2, 724.). It suggests that, in artificial water oxidation catalysts, 'smart' removal of protons from the catalytic site may be also an issue when aiming at fast and efficient water oxidation. We believe that it is possible to tune the catalytic activity by tuning the pKa of additional uncoordinated ligands.

Figure for Reviewer #2. Cyclic voltammetry curves of NiCo composites with terephthalate with different functional groups as proton relay measured in 1.0 M KOH.

Actually, we have tried to change the pKa of uncoordinate carboxylates for NiCo composites by immobilizing the structure of terephthalate with electron-withdrawing and electron-donating groups, respectively, preliminary results are shown in above Figure. This is an ongoing work and final results and discussions will be published in the future.

Reviewer 3.

Reviewer's Comments and Questions:

1. The paper of Li et al. reports on an iron/nickel terephthalate coordination polymer deposited on nickel foam (NiFeCP/NF) to catalyze the anodic oxygen evolution reaction in alkaline media. Proposed in this work catalyst requires an overpotential of 188 mV at 10 mA cm² and exhibits a Tafel slope of 29 mV dec⁻¹ and, thus, shows better performance compared to the chosen benchmark material NiFe LDH. However, NiFe-based catalysts with even lower Tafel slopes were reported in literature (e.g. catalyst with Tafel slope of 15 mV dec⁻¹ in *J. Phys. Chem. C* 2008, 112, 3655-3666). Overall approach of incorporation of a catalyst in a polymer matrix can be interesting, but there are concerns regarding the stability of such polymer on the long run under the OER conditions. Polymers are typically unstable in the alkaline media, and their stability is drastically decreasing at high anodic potentials because the functional groups providing conductivity are prone to be attacked by the OH radicals. This challenge is very difficult to overcome and this is hindering development of membrane for alkaline water electrolysis in particular. There is no evidence in the current manuscript that the polymer matrix is stable during long term operation. Moreover, the HAADF-STEM images presented in Figure 2 suggest decrease in Carbon content after the electrolysis which indicates the degradation of the polymer. The data on electrochemical stability of the polymer itself is crucial, still it is not provided in the manuscript. The main question is what would be the gain in incorporating the catalyst into potentially unstable polymer matrix if already reported in the literature NiFe-based catalysts have comparable or even better performance? I doubt that without this important information the manuscript can influence the thinking in the field of electrocatalysis, as one would expect from the paper published in Nature Communications.

Response: We have studied all your comments, questions and recommended literature carefully, and tried our best to answer your questions and revise the manuscript accordingly.

First of all, please allow us to explain the purpose and consideration of this work. In the past few decades, extensive research on first row transition metal based materials has been reported as heterogeneous OER catalysts, and significant efforts have been expended on morphology control, element doping etc aiming at increasing the apparent activity of such catalysts. Meanwhile, most of OER catalysts are reported undergoing concerted proton-coupled electron transfer (c-PET) processes, and the rate of proton transfer plays an important role for OER. So far, however, research efforts have not been focused on how to accelerate the rate of proton transfer for heterogeneous OER catalysts.

As known that, the mass of proton and electron are 1.67×10^{-27} kg and 9.11×10^{-31} kg, respectively. Because the proton has a much larger mass than the electron, proton transfer is considered to be much slower and which will control the reaction rate of a PCET reaction (J. Am. Chem. Soc. 2011, 133, 13224; Chem. Rev. 2010, 110, 6939). Understanding of the underlying mechanism of proton-coupled interfacial electron transfer process in the rate-determining step for the current strategies of catalyst design is obviously lacking of this important part. Our work is exactly focusing on this puzzle, therefore, we believe that this work is important and suitable for Nature Communications.

The most important part of this work is the postulated secondary coordination sphere effects (uncoordinated carboxylates) that could be contributing to accelerate proton transfer in the rate-determining step and enhance the activity of OER catalysts.

We completely agree with you that some NiFe-based catalysts may display even more outstanding activities, but not every catalyst can be used as the reference to get important kinetic information. In our work, we claimed that NiFeCP shows better performance compared to the benchmark material NiFe LDH, because of the much lower overpotential and the much higher ECSA normalized current density, not just due to the low Tafel slope. Through the comparison between NiFeCP (uncoordinated carboxylate containing catalyst) and NiFe LDH (carboxylate-free catalyst), we could reveal the function of uncoordinated carboxylate groups for concerted proton-electron transfer pathways, and this is what we want to express in our work.

We have seriously studied the literature you pointed out (J. Phys. Chem. C 2008, 112, 3655-3666, where “the NiFe oxide catalyst achieved a nearly ideal anodic electron-transfer coefficient, $\alpha_a=0.0082$ 14.8 mV dec⁻¹ in a 1 M KOH solution”). For an OER catalyst, only the rate determining step is a sequential proton-electron transfer (s-PET) step, at the same time, proton transfer is not the involved in this step, the theoretical value of the Tafel slope could be as small as $2.303 \times 2RT/7F$ (17 mV dec⁻¹) or $2.303 \times RT/4F$ (15 mV dec⁻¹) (The Journal of Chemical Physics 1956, 24, 817; Phys. Chem. Chem. Phys. 2011, 13, 21530; Phys. Chem. Chem. Phys. 2013, 15, 13737; Sci. Rep. 2015, 5, 13801). In practice, such kind of OER catalysts are rare, because proton has a much larger mass than that of electron, it's difficult to make the rate determining step to be irrelevant with proton transfer. The literature you mentioned (with Tafel slope of 15 mV dec⁻¹), which may undergo sequential proton-electron transfer pathway. Our work is focusing on how to accelerate the proton transfer for concerted proton-electron transfers of OER, sequential proton-electron transfer pathway is not the focus in our manuscript.

About the stability: Recently, some transition MOFs with excellent stability have been reported, such as NiCo-UMOFNs (Nature Energy 2016, 1, 16184); NiFe MOF (MIL-53)

(Adv. Energy Mater. 2018, 8, 1800584); Ni/Fe bimetal two-dimensional (2D) ultrathin MOF (Nano Energy 2018, 44, 345). In these cases, the post characterizations have proved that the chemical structures are stable. According to the recommendations of IUPAC (DOI: <https://doi.org/10.1351/PAC-REC-12-11-20>), MOFs can be classified as kinds of coordination polymer, as the terephthalate ligand used in our work is the same as in literature. The post characterizations have proved that there are still uncoordinated carboxylate groups maintained after long time electrolysis in our system. The structural changes are the results of excessive uncoordinated carboxylate groups existing in the as prepared NiFeCP. We prepared the NiFeCP as an OER catalyst by a fast electrochemical deposition process, in which both coordinated and uncoordinated carboxylate groups were introduced to the NiFeCP film, which means that the excessive uncoordinated carboxylate groups from terephthalate were introduced into NiFeCP on purpose in our work. "Excessive terephthalates" are expected to be removed during activation process. We are very sorry to make you confused due to the unclear description in the original manuscript. We have made a clearer description in the revised supporting information as following: "Before water oxidation measurements, the as prepared **NiFeCP/NF** electrodes were activated at 50 mA cm⁻² current density in 1.0 M KOH for 10 mins to remove the excessive terephthalates in **NiFeCP** introduced by the fast electrochemical deposition process. The activated **NiFeCP/NF** electrodes were rinsed with water, dried in air, and then water oxidation activities were measured the in fresh electrolytes."

About "the HAADF-STEM images presented in Figure 2 suggest decrease in Carbon content" you mentioned. First, as long as carbon content in the NiFeCP particle could be observed after OER on HAADF-STEM images, together with FTIR, Raman and XPS, which can prove that carboxylates still existed in NiFeCP. Second, in our supporting information, we have described the measurement of TEM, samples "were added drop-wise onto a carbon-coated copper grid". Because of that the carbon-coated copper grid is the substrate for TEM measurement, which will affect the quantitative analysis, therefore, we didn't discuss the quantity changes of carbon content from HAADF-STEM.

Question 2

Another important point that is missing in the manuscript is comparison of the electrochemically active surface area before the anodic scan and after it. From the XRD data presented in Figure S5, it becomes clear that the OER leads to amorphization of the

catalytic surface, which will affect the ECSA. Considering that polarization curves presented in Fig. 2 are taken till relatively high current densities the amorphization is probably ongoing while the polarization curve is being recorded and related to it increase of ECSA may significantly affect the measured reactivity.

Response: According to you suggestions, the ECSA after OER, has been measured, as shown in Figures S3 and S5. The related discussions have been added to the revised manuscript.

Figure S3 and S5. Cyclic voltammetry (CV) curves of (a) NiFeCP/NF after OER and (c) activated NiFeCP/NF in 1 M KOH with different scan rates at selected potential range; (b) and (d) the corresponding capacitance Δj ($|j_{charge} - j_{discharge}|$) versus the scan rates.

Question 3

The XPS fitting model does not really match the experimental curve and some of important features are not considered in Figure 3. It is well known that anodic polarization of Ni-Fe based materials leads to formation of oxyhydroxides. However, the peak corresponding to the OH groups is missing in the XPS fitting model of O 1s level presented in Figure 3b. This should be corrected, since hydroxy species are included in the spectra of Ni and Fe in Figure 3. Also the fitting model doesn't include contribution of

several important features, e.g. fitted line doesn't include small shoulder at ca 529 eV in Figure 3b after OER. The additional component at 583 eV is missing in the fitting model of Ni spectrum.

Response: XPS fittings have been improved according to you suggestions, as shown in Figure 3.

Figure 3. High-resolution XPS spectra of a) C 1s, b) O 1s, c) Fe 2p, and d) Ni 2p for as prepared NiFeCP/NF and NiFeCP/NF electrode after OER test.

Question 4

What was the collection efficiency in GC measurements to estimate the efficiency of the OER? Ni and Fe are both stable towards dissolution under anodic polarization in the alkaline medium. What electrochemical process is responsible for the rest 4% of the current? Polymer oxidation/degradation?

Response: We have re-measured the Faradaic Efficiency of NiFeCP in a small volume electrolysis cell, the collection efficiencies are $98.4 \pm 0.6\%$. The Faradaic Efficiency of NiFeLDH was also measured, a Faradaic Efficiency of $97.8 \pm 1.7\%$ has been obtained.

Figure S6. The Faradaic efficiency of (a) **NiFeCP/NF** and (b) **NiFe LDH/NF** for OER in 1.0 M KOH H₂O. Comparison of the amount O₂ of the theoretically calculated and experimentally measured gas versus quantity of electric charge for water splitting catalyzed by the **NiFeCP/NF** and **NiFe LDH/NF** at a current density of $j = 10 \text{ mA cm}^{-2}$. The quantitative yield of **NiFeCP/NF** and **NiFe LDH/NF** was $98.4 \pm 0.6\%$ and $97.8 \pm 1.4\%$, respectively.

We have even measured the Faradaic Efficiency of NiFeCP and NiFeLDH in the deuterated electrolyte (1.0 M NaOD in D₂O). The results indicate that the accumulated charge for both NiFeCP and NiFeLDH electrodes can be almost quantitatively consumed for OER.

Figure S14. The Faradaic efficiency of (a) **NiFeCP/NF** and (b) **NiFe LDH/NF** for OER in a 1.0 M NaOD D₂O solution. Comparison of the amount O₂ of the theoretically calculated and experimentally measured gas versus quantity of electric charge for water splitting catalyzed by the **NiFeCP/NF** and **NiFe LDH/NF** at $j = 10 \text{ mA cm}^{-2}$.

REVIEWERS' COMMENTS:

Reviewer #1 (Remarks to the Author):

The authors report a NiFe coordination polymer material (NiFeCP) which when adsorbed on Ni foam electrodes shows promising activity and selectivity for the OER. The authors compare this material to a NiFeLDH/NF material without a coordination polymer, and the NiFeCP/NF shows improved activity for OER. The authors show that NiFeLDH/NF has a KIE ~ 3 but NiFeCP/NF has a KIE ~ 2.1 . Proton inventory studies show a non-linear dependence on electrolyte deuteration. Both KIE suggest a rate-determining proton transfer step, but the smaller KIE for NiFeCP/NF was used to suggest there is a secondary-coordination sphere effect from uncoordinated terephthalate groups in the polymer that help OER at the NiFeCP/NF material and/or a proton-relay effect for proton delivery as shown in Scheme 2.

This is a comprehensive study that has significantly strengthened based on the authors' responses to reviewer comments. I believe this is an exciting example of using the second coordination sphere to tune activity at active sites in a solid-state heterogeneous catalytic material that will be of broad interest in the field. The authors have adequately addressed reviewer concerns, and I support this manuscript's publication without further revision.

Reviewer #2 (Remarks to the Author):

The revised manuscript has been significantly improved. All issues that reviewers are concerned have also been clarified and the experiment has been further refined as well. I suggest this work to be accepted for publication.

Reviewer #3 (Remarks to the Author):

The authors have significantly improved the manuscript and addressed all the questions the referee asked. The recommendation is to accept the manuscript.